

# Clinical performance of zirconium implants compared to titanium implants: a systematic review and meta-analysis of randomized controlled trials

Chengchen Duan[1,*], Li Ye[2,*], Mengyun Zhang[1], Lei Yang[1], Chunjie Li[3], Jian Pan[2], Yingying Wu[4] and Yubin Cao[2]

[1] State Key Laboratory of Oral Diseases, National Clinical Research Center for Oral Diseases, West China College of Stomatology, Sichuan University, Chengdu, China

[2] State Key Laboratory of Oral Diseases, National Clinical Research Center for Oral Diseases, Department of Oral and Maxillofacial Surgery, West China Hospital of Stomatology, Sichuan University, Chengdu, China

[3] State Key Laboratory of Oral Diseases, National Clinical Research Center for Oral Diseases, Department of Head and Neck Oncology, West China Hospital of Stomatology, Sichuan University, Chengdu, China

[4] State Key Laboratory of Oral Diseases, National Clinical Research Center for Oral Diseases, Department of Oral implantology, West China Hospital of Stomatology, Sichuan University, Chengdu, China

[*] These authors contributed equally to this work.

Corresponding authors
Yingying Wu, yywdentist@163.com
Yubin Cao, yubin.cao@qq.com, yubin.cao@scu.edu.cn

## ABSTRACT

**Purpose.** To quantitatively assess and compare the clinical outcomes, including survival rate, success rate, and peri-implant indices of titanium and zirconium implants in randomized controlled trials.

**Methods.** The electronic databases searched included the Cochrane Central Register of Controlled Trials (CENTRAL), Medline *via* Ovid, EMBASE, and Web of Science. Randomized controlled trials (RCTs) that reported the effects of zirconium implants on primary outcomes, such as survival rate, success rate, marginal bone loss (MBL), and probing pocket depth (PPD), compared to titanium implants were included in this review. Two reviewers independently screened and selected the records, assessed their quality, and extracted the data from the included studies.

**Results.** A total of four studies from six publications reviewed were included. Two of the comparative studies were assessed at minimal risk of bias. Zirconium implants may have a lower survival rate (risk ratio (RR) = 0.91, CI [0.82–1.02], $P = 0.100$, $I^2 = 0\%$) and a significantly lower success rate than titanium implants (RR = 0.87, CI [0.78–0.98], $P = 0.030$, $I^2 = 0\%$). In addition, there was no difference between the titanium and zirconium implants in terms of MBL, PPD, bleeding on probing (BOP), plaque index (PI), and pink esthetic score (PES) (for MBL, MD = 0.25, CI [0.02–0.49], $P = 0.033$, $I^2 = 0\%$; for PPD, MD = $-0.07$, CI [$-0.19$–0.05], $P = 0.250$, $I^2 = 31\%$).

**Conclusion.** Zirconium implants may have higher failure rates due to their mechanical weakness. Zirconium implants should be strictly assessed before they enter the market. Further studies are required to confirm these findings.

## INTRODUCTION

Dental implants have become a reliable treatment option for restoring missing teeth because of their ability to restore chewing, speech, and aesthetic functions effectively (*Buser et al., 2012*; *Buser, Sennerby & Bruyn, 2017*; *Osman & Swain, 2015*; *Pjetursson et al., 2012*). Titanium (Ti) is the golden standard material for fabricating dental implants due to its superior biocompatibility, mechanical properties, and promising long-term survival rates (*Cao et al., 2019*; *Haugen & Chen, 2022*; *Najeeb et al., 2016*; *Rokaya et al., 2022*; *Sivaraman et al., 2018*). However, the mechanical and elastic modulus differences between bone and titanium can cause bone loss and "stress shielding" (*Khurshid et al., 2020*). Moreover, Ti implants' grayish appearance may result in peri-implant mucosa discoloration, particularly in patients with thin gingival phenotypes (*Wang et al., 2021*). In addition, Ti ion dissolution related to implant corrosion could cause plaque formation and corrosion product accumulation, which are correlated with peri-implantitis and allergic reactions (*Noronha Oliveira et al., 2018*; *Noumbissi, Scarano & Gupta, 2019*).

Ceramic implants are light-transmissive, and their colors are closer to those of natural teeth, resulting in excellent aesthetic performance. Histological observations also showed that zirconium (Zr) implants had similar biocompatibility and osseointegration to Ti implants (*Chacun et al., 2021*). A previous study reported that the early fracture of Zr implants, caused by poor mechanical properties, may limit its clinical application (*Sivaraman et al., 2018*). Recently, the flaws in the mechanical properties and insufficient strength in Zr were addressed by adding stabilizers such as yttrium oxide, cerium oxide, and magnesium oxide (*Burkhardt et al., 2021*; *Khurshid et al., 2020*; *Pardun et al., 2015*; *Shan et al., 2022*). Although Zr implants demonstrated some stability and resistance to fracture, it remained controversial whether they could be widely used in the clinical practice.

A few systematic reviews and meta-analyses comparing the properties of Zr and Ti implants were previously conducted. A systematic review of *in vivo* experiments found no differences in bone-implant contact (BIC) and removal torque between Zr and Ti implants (*Manzano, Herrero & Montero, 2014*). A meta-analysis conducted in 2017 did not evaluate Zr and Ti implants in randomized controlled trials (RCTs) with a comparable baseline and has been outdated (*Elnayef et al., 2017*). Furthermore, a recent systematic review did not quantitively analyze the differences between Zr and Ti implants (*Fernandes et al., 2022*). As such, a novel systematic review and meta-analysis of RCTs is necessary to provide up-to-date evidence-based recommendations. Therefore, this systematic review and meta-analysis aimed to quantitatively assess and compare the clinical outcomes, including survival rate, success rate, and peri-implant indices of Ti and Zr implants in RCTs.

## MATERIALS & METHODS

This systematic review and meta-analysis was conducted in accordance with the Preferred Reporting Items for Systematic Reviews and Meta-Analyses (PRISMA) statement (*Page et al., 2021*). This study has been prospectively registered on the PROSPERO website (CRD42022354985). Due to the delayed protocol submission to PROSPERO, we have

started the formal screening of search results against eligibility criteria before the submission to PROSPERO.

## PICOS question and eligibility criteria

Articles were included in this systematic review if they met the following inclusion criteria formulated using the PICOS question: participants (P), patients with one or more missing teeth who received one or more dental implants; intervention (I), veneered or monolithic Zr dental implants; comparison (C), traditional Ti implants. We did not restrict the type of abutment or prosthetic restoration. The primary outcomes (O) were survival rate, success rate, marginal bone loss (MBL), and probing pocket depth (PPD); secondary outcomes included plaque index (PI), bleeding on probing (BOP), pink aesthetic score (PES), and reasons for failure. The study design (S) of interest was randomized controlled trials (RCT).

## Search strategy

Cochrane Central Register of Controlled Trials (CENTRAL), Medline *via* Ovid, EMBASE, and Web of Science were searched from inception to August 2022 by two authors (C. D. and L. Y.). The strings of the specific search strategies for these databases are presented in Table S1. The reference lists of the included studies were manually checked to identify potentially relevant studies. The articles obtained were imported into EndNote X9 (Thomson Reuters, Philadelphia, PA, USA) for screening.

## Study selection

The titles and abstracts of all articles identified from the electronic database search were independently reviewed by two researchers (C. D. and L. Y). Full texts of potentially relevant articles were retrieved and further evaluated. Disagreements were resolved through discussion or by consulting an arbitrator (Y. C.). The reasons for exclusion were recorded for all the full texts. The study centers, ethical committee approval number, and full author lists of the selected studies were double-checked to detect replicated publications that reported on the same trial before being included.

## Data extraction

Data extraction was performed independently by two researchers (C. D. and L. Y.) using a pre-designed form. When necessary, the corresponding authors of included studies were contacted to obtain additional information. The following data were extracted: authors' names, publication date, country of origin, participants' demographic characteristics, follow-up, intervention type (abutments, implants, or prostheses), and outcomes. Disagreements were settled through discussion with an arbitrator (Y. C.). Success was defined as the absence of biological or technical implant complications. A functional implant that still failed to meet the success criteria was considered to have survived. Failure occurred when the implant was removed for any reason.

## Risk of bias assessment

The bias of the included RCTs was evaluated using the Cochrane Risk of Bias Assessment Tool *via* RevMan software (version 5.3, The Cochrane Collaboration, Copenhagen, Denmark) (*Cochrane Training, 2023*).
Two reviewers (C. D. and L. Y.) judged the risks as either "High", "Low", or "Unclear" based on seven domains: random sequence generation (selection bias), allocation concealment (selection bias), blinding of participants and personnel (performance bias), blinding of outcome assessment (detection bias), incomplete outcome data (attrition bias), selective reporting (reporting bias), and other bias. Disagreements were resolved through discussion between the two reviewers (C. D. and L. Y.). Any unresolved divergence was settled by an arbitrator (Y. C.).

## Statistical analysis

The meta-analysis was performed using Review Manager 5.4.1 (The Cochrane Collaboration, London, UK) and STATA 15.0 (StataCorp LLC, College Station, TX, USA) software. Dichotomous variables were described as a percentage of the total number of events. The difference was calculated using the risk ratio (RR). Continuous variables were presented as mean and standard deviation and differences were evaluated using pooled mean difference (MD). The 95% confidence interval (CI) and $P$ value were also calculated.

A Q-test based on the Cochran Chi-square test and Higgins I-squared statistic ($I^2$) was used to detect statistical heterogeneity in this meta-analysis. A fixed-effects model was selected when $I^2$ was < 50%. When methodological heterogeneity was present, a sub-group analysis was performed. The subgroups were divided according to the follow-up duration or mucositis status. The stability of the results was evaluated using sensitivity analysis, which was performed by excluding one study from the meta-analysis and overserving whether the pooled results were significantly changed. If more than six studies were combined in the meta-analysis, funnel plots were used to assess publication bias. Qualitative descriptions were presented when data could not be pooled.

### Certainty of evidence

The GRADE pro system (https://www.gradepro.org/) was used to evaluate the certainty of the evidence for all the outcomes (*Schünemann et al., 2008*). The included studies were RCTs with the highest pre-determined certainty of evidence. The certainty of evidence could be described as high, moderate, low, or 'very low' depending on the following five domains: risk of bias, inconsistency of effect, indirectness, imprecision, and publication bias (*Guyatt et al., 2008*).

## RESULTS

### Study selection

A total of 4,878 articles were identified through electronic and manual searches and 1,746 duplicates were excluded *via* EndNote X9. After initial screening by titles and abstracts, 3,107 records were excluded from the initial screening. We reviewed the full texts of the remaining 29 articles and finally included four RCTs with six records (*Bienz et al., 2021*; *Koller et al., 2020*; *Osman et al., 2014b*; *Payer et al., 2015*; *Ruiz Henao et al., 2021*; *Siddiqi et al., 2015*). A flow diagram of this study is presented in Fig. 1A.

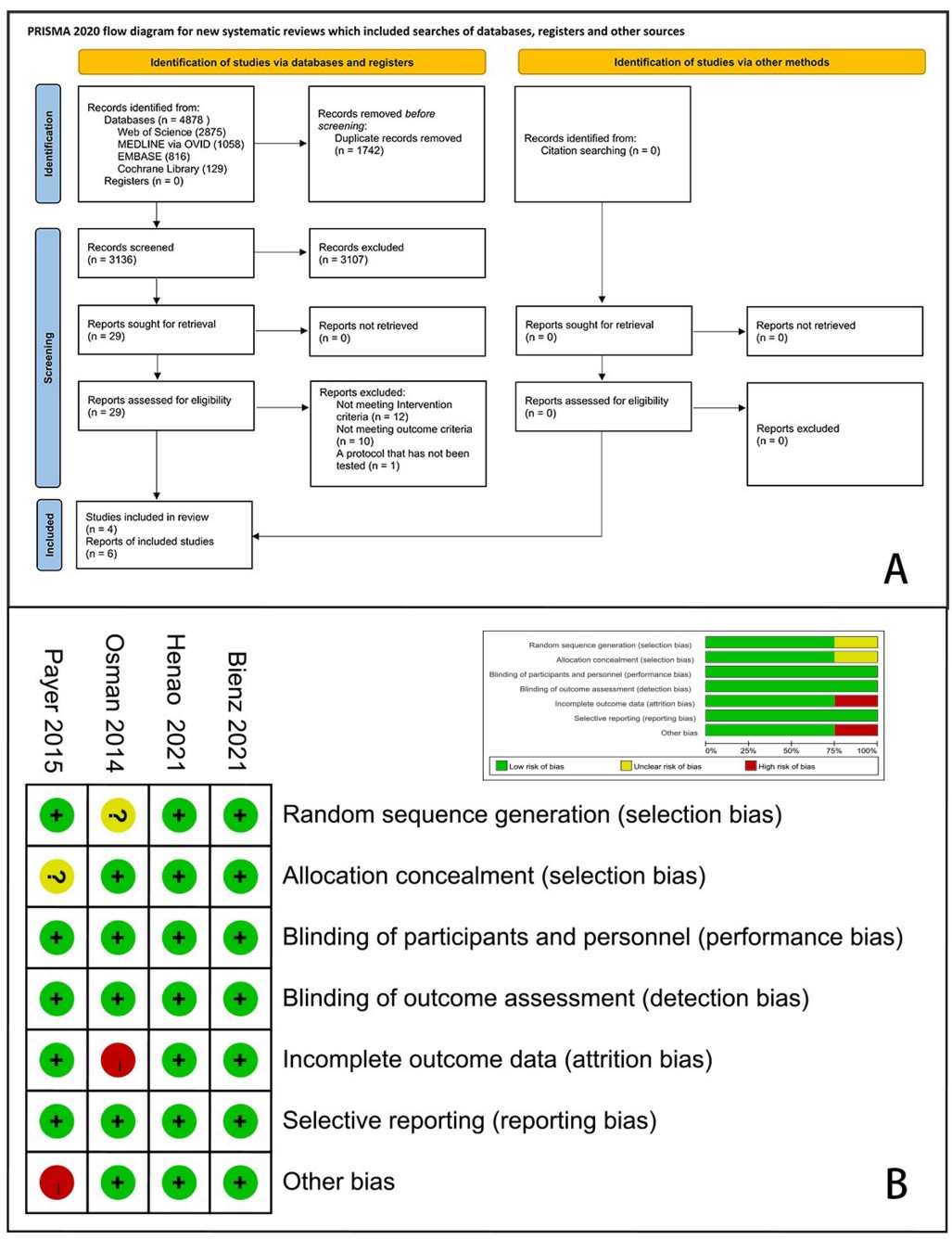

**Figure 1** **PRISMA flow chart for study selection and risk of bias of the included RCTs.** (A) Flow chart; (B) risk of bias plot.

## Study characteristics

The basic characteristics of the included studies are presented in Table 1. These studies were published between 2014 and 2021, with a mean follow-up period of 23 months. A total of 109 participants with 248 implants were evaluated for comparison purposes. All implants

in the RCTs were placed using the delayed approach. *Payer et al. (2015)* used two-piece implants (*Koller et al., 2020*), while the rest of the studies used monotype or one-piece implants. In addition, *Osman et al. (2014b)* rehabilitated completely edentulous patients (*Siddiqi et al., 2015*), whereas other researchers have only partially restored edentulous dentition. Moreover, only *Bienz et al. (2021)* compared Zr and Ti implants under healthy and experimental mucositis conditions.

## Risk of bias

The quality assessment and risk of bias of the included RCTs are shown in Fig. 1B. Two studies were assessed as having an elevated risk of bias due to excessive loss during follow-up (*Osman et al., 2014b*) or baseline incomparability between the control and experimental groups (*Payer et al., 2015*). Moreover, we determined that the risk of bias was unclear in this study. Other studies were assessed to have a minimal risk of bias.

## Data synthesis

(1) Survival rate: Four articles provided survival data and were included in this meta-analysis (*Koller et al., 2020*; *Osman et al., 2014b*; *Payer et al., 2015*; *Ruiz Henao et al., 2021*). No significant difference was found between Zr and Ti in the survival rate (RR = 0.91, CI [0.82–1.02], $P = 0.100$, $I^2 = 0\%$) (Fig. 2A). No significant heterogeneity was observed in the subgroups with different follow-up durations.

A total of 24 Zr implants failed for several reasons, including osseointegration failure ($n = 18$), fracture ($n = 3$), adequate torque verification ($n = 2$), and excessive occlusal loading ($n = 1$). However, only 11 Ti implants failed because of osseointegration failure ($n = 10$) and peri-implantitis ($n = 1$).

(2) Success rate: Four articles provided survival data and were included in the meta-analysis (*Koller et al., 2020*; *Osman et al., 2014b*; *Payer et al., 2015*; *Ruiz Henao et al., 2021*). A statistically significant difference was found in the survival rate of different implant materials (RR = 0.87, CI [0.78–0.98], $P = 0.030$, $I^2 = 0\%$) (Fig. 2B). There was no significant heterogeneity between the subgroups at the different follow-up times.

The reasons for the lack of success in the Ti group are described above, while there are three more implants in the Zr group owing to their deep placement.

(3) MBL: Three studies with four reports were included in the meta-analysis (*Koller et al., 2020*; *Osman et al., 2014b*; *Payer et al., 2015*; *Ruiz Henao et al., 2021*). One study reported pre- and post-changes in MBL, while others only reported MBL at the end of the follow-up period. *Osman et al. (2014b)* favored Ti rather than Zr implants because of decreased MBL (Ti: 0.18 ± 0.47 mm, Zr: 0.42 ± 0.40 mm, $P = 0.024$). A meta-analysis of other studies showed a significantly higher difference in the MBL of Zr implant materials compared to Ti implant materials (MD = 0.25, CI [0.02–0.49], $P = 0.033$, $I^2 = 0\%$) (Fig. 2C). There was no heterogeneity in the subgroup follow-up durations.

(4) PPD: Three articles provided survival data and were included in this meta-analysis (*Bienz et al., 2021*; *Ruiz Henao et al., 2021*; *Siddiqi et al., 2015*). The two subgroups were divided according to mucosal conditions (normal *versus* experimental mucositis). The results showed no significant difference between Zr and Ti in terms of PPD (MD = −0.07,

Duan et al. (2023), *PeerJ*, DOI 10.7717/peerj.15010

**Table 1  The characteristics of the included studies in the final meta-analysis (four RCTs with six records).**

| Study | Location | Participants | | Implant site | | Follow up (month) | Drop out (participants, implants) | Interventions (implant, abutment, prosthesis) | Outcomes |
|---|---|---|---|---|---|---|---|---|---|
| | | Age (mean, range) | Gender (F/M) | Anterior/ Posterior | Maxilla/ Mandible | | | | |
| *Ruiz Henao et al. (2021)* | America | 55.00 | 16/14 | Gp1: 16/0 Gp2: 14/0 | NS | 12 | 0 | Implants: Gp1 ($n = 16$): ZLA Ceramic monotype implant (Straumann) Gp2 ($n = 14$): SLA Titanium implant (Straumann) Abutments: NS Prosthesis: polyether impression material (Impregum), crowns | PES, PPD, MBL |
| *Bienz et al. (2021)* | Switzerland | 55.4, 45.6-65.3 | 14/8 | Gp1: 0/40 Gp2: 0/40 | Gp1: 14/26 Gp2: 14/26 | 0.75 | 0 | Implants: Gp1 ($n = 20$): yttrium-stabilized zirconium dioxide (Institut Straumann AG) Gp2 ($n = 20$): titanium (Institut Straumann AG) Abutments: NA Prosthesis: NA | PPD, BOP |
| *Payer et al. (2015)* | Austria | 46, 24~77 | 9/13 | Gp1: 3/13 Gp2: 2/13 | Gp1: 3/13 Gp2: 4/11 | 24 | 0 | Implants: Gp1 ($n = 16$): yttria-stabilized zirconium (Ziterion) Gp2 ($n = 15$): two-piece titanium (Ziterion) Abutments: Gp1 ($n = 16$): Zirconium abutments (Ziterion) Gp2 ($n = 15$):titanium abutments (Ziterion) Prosthesis: all-ceramic crowns | PI, BOP, PES, MBL |

**Table 1** (*continued*)

| Study | Location | Participants | | Implant site | | Follow up (month) | Drop out (participants, implants) | Interventions (implant, abutment, prosthesis) | Outcomes |
|---|---|---|---|---|---|---|---|---|---|
| | | Age (mean, range) | Gender (F/M) | Anterior/ Posterior | Maxilla/ Mandible | | | | |
| *Koller et al. (2020)* | | | | | | 80.9 ± 5.5 | 1, 3 | | |
| *Osman et al. (2014b)* and *Siddiqi et al. (2015)* | Egypt | 62, 46~80 | 4/15 | NS | NS | 12 | 5, 35 | Implants: Gp1 ($n = 73$): Zirconium (Southern Implants) Gp2 ($n = 56$): titanium (Southern Implants) Abutments: NS Prosthesis: overdentures | MBL, PI, PPD |

**Notes.**

Gp, group; RCT, randomized controlled trial; NS, not stated; NA, not applicable; MBL, marginal bone loss; MB level, marginal bone level; PPD, pocket probing depth; BOP, bleeding on probing; PI, plaque index; PES, pink esthetic score.

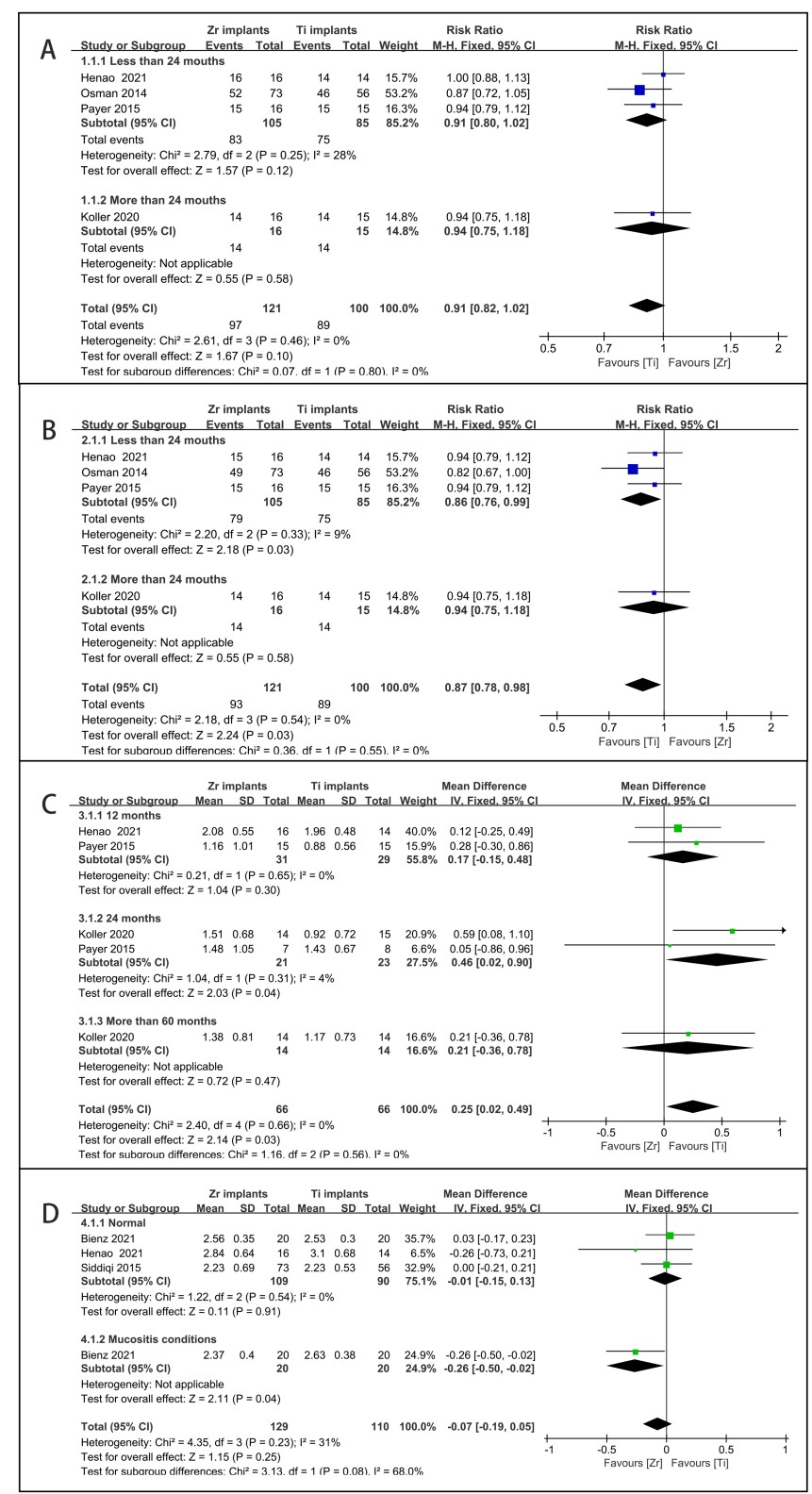

**Figure 2   Forest plot presenting the results of meta-analyses to compare clinical performances of Zr and Ti implants.** (A) Survival rate; (B) Success rate; (C) MBL; (D) PPD.

CI [$-0.19$–$0.05$], $P = 0.250$, $I^2 = 31.3$%) (Fig. 2D). Mucosal conditions may explain the source of heterogeneity.

(5) BOP: *Bienz et al. (2021)* and *Koller et al. (2020)* reported no significant differences in BOP between Zr and Ti implants under healthy conditions, but the results were more favorable for Zr implants under experimental mucositis conditions, with a significantly lower BOP (Zr: $21.7 \pm 23.6$%, Ti: $32.5 \pm 27.8$%, $P < 0.001$).

(6) PI: *Bienz et al. (2021)* noted that the results were more favorable for Zr implants due to a lower PI, with a significant difference under mucositis conditions (Zr: $68.3 \pm 31.9$%, Ti: $75.0 \pm 29.4$%, $P < 0.001$), while *Siddiqi et al. (2015)* and *Koller et al. (2020)* concluded that PI was not significantly different between Zr and Ti implants.

(7) PES: Two studies with three reports that assessed PES suggested a positive performance of Zr implants to improve aesthetic outcomes (*Koller et al., 2020*; *Payer et al., 2015*; *Ruiz Henao et al., 2021*). In the gingival margin position, the contour of the labial surface and the color and texture of this peri-implant tissue, the results were slightly better when a Zr implant was placed (*Ruiz Henao et al., 2021*). Zr implants showed a significantly higher difference ($P = 0.004$) compared to Ti implants at baseline and the 6-, 12-, 18-, and 24-months follow-up, but this difference was not observed at the 80-month follow-up ($P = 0.428$) (*Koller et al., 2020*; *Payer et al., 2015*).

### Sensitivity analysis and publication bias

All sensitivity analyses showed the robustness of the meta-analysis (Figs. S1–S4). Publication bias was not assessed for all outcomes due to the limited number of included studies.

### Certainty of evidence

The GRADE certainty of evidence ranged from moderate to very low (Table S2). The success rates were downgraded to moderate, the survival rate, PPD, BOP, and PES were low, and the MBL and PI were very low. The grade of the evidence was lowered by limitations in study quality, inconsistency of results, or imprecision of the study design.

## DISCUSSION

This systematic review compared the clinical performance of Zr and Ti dental implants utilized in RCTs. Results from four RCTs involving 248 implants were pooled, and the success rate of Zr implants was significantly lower than that of Ti implants, while the survival rate showed a similar tendency, although not significant. The results were consistent with those of other reports on survival or success rates (*Becker et al., 2017*; *Cannizzaro et al., 2010*; *Cionca, Müller & Mombelli, 2015*; *Spies et al., 2019*; *Spies et al., 2017*; *Spies, Stampf & Kohal, 2015*; *Spies et al., 2018*). The survival rates of Zr implants were estimated to be 92.77%, 98.58%, and 86.38% at 1-, 2-, and 5-year follow-up, respectively (Table S3), while the success rate was 84.89%, 80.78%, and 66.38%, respectively (Table S4).

The expectations for dental implants include a 5-year success rate of $>85$% for maxillary implants and $>95$% for mandibular implants. However, Zr implants may not achieve the desired success rate. Chipping and occlusal roughness were the main technical complications of long-term restorations. Even for the same surgeon, the success rate can

range from 38–91%. Hence, we consider that the manufacturer of Zr implants may be the main contributor to the extreme heterogeneity in survival and success rates among RCTs or cohorts. We recommend that Zr implants should be subjected to stricter assessments before being distributed for clinical use. We also analyzed other factors that may contribute to the failure of Zr implants (Table S5), which revealed that posterior tooth sites were correlated with higher prosthetic failure. Therefore, we advise against the use of Zr implants in the molar region.

In the pooled analysis of MBL, the results favored the use of Ti implants. However, it must be noted that the MBL comparison between Zr and Ti implants revealed no statistical difference in all included studies, except for the study by *Osman et al. (2014b)* that showed an increased bone loss and higher fracture rate in Zr implants for overdentures. Additionally, the success rate was also significantly lower in the Zr group. This significant difference could be explained by the selection of participants and implant diameter. The occlusal load required for overdentures in edentulous patients may be far larger than that in single crowns, while inadequate mechanical strength has always been a flaw in Zr implants. An interview revealed that the color of teeth and denture flanges representing gingival tissues was more important to the patients than the implants' color from an aesthetic point of view (*Osman et al., 2014a*). Therefore, overdentures on Zr implants have additional risks but no benefits, which should not be recommended in any clinical setting except in cases where the patient has an allergy to titanium.

Furthermore, this review may be the first to reveal that the PPD of Ti and Zr implants is comparable and significant under normal and experimental mucositis conditions, respectively. A recent systematic review failed to draw a definitive conclusion owing to the limitation in sample sizes of the included studies (*Fernandes et al., 2022*). Under healthy conditions, a study revealed that the PPD was significantly greater around Ti implants than around teeth in the same patient (mean PPD difference = 0.8 mm, SD = 0.1 mm, $P < 0.001$) after 3-year follow-up (*Johnson & Persson, 2000*), whereas the PPD around the Zr implants statistically significantly lower than around the control teeth at a 3-year follow-up time ($r = 0.56$, $P < 0.003$) was observed in another trial (*Brüll, Van Winkelhoff & Cune, 2014*). However, the clinical significance of these differences is debatable due to the incomparable baseline data and small variance values, respectively. Under mucositis conditions, a significantly higher PPD around Ti implants than around Zr implants was reported after 5 months in a cohort study ($P = 0.003$) (*Clever et al., 2019*), which is consistent with findings from the included RCT (*Bienz et al., 2021*). This could be due to the significant decrease in periodontal pathogen adhesion on Zr implants, which reduces plaque accumulation and inflammation (*Roehling et al., 2017*). Therefore, Zr and Ti implants can both achieve healthy and stable soft tissues, but Zr implants may compare favorably with the mucogingival findings reported on mucositis conditions involving Ti implants.

This systematic review found contradictory results regarding PI between the two groups. Two studies did not support any difference in PI, which is inconsistent with the results of preclinical studies. Higher numbers of microorganisms and counts of pathogenic species on the Ti implants were associated with a significantly greater roughness compared to Zr

implants (*Do Nascimento et al., 2013*). However, the surface modification methods such as acid-etched implants and sandblasted implants contribute to increased surface roughness and enhanced osseointegration in Ti implants compared to Zr implants (*Kniha et al., 2021*; *Sivaraman et al., 2018*). Compared to surface roughness, patient compliance, including plaque control and dental follow-up, may be more important in maintaining oral hygiene and preventing peri-implantitis (*Cortellini et al., 2019*). According to the literature, we consider that patients with either Zr or Ti implants could avoid bacterial cumulation and implant loss due to marginal inflammation by optimal compliance to oral self-care and a supportive periodontal therapy program.

The PES results affirm the promising short- and medium-term aesthetic outcomes observed in studies examining the behavior of soft tissues around Zr implants (*Kniha et al., 2019*; *Sadowsky, 2020*). Nevertheless, the difference between Ti and Zr implants may not be aesthetically visible because the PES of both implants was above the threshold of clinical acceptance set at a score of six. The aesthetic outcomes seem to be more related to the thickness of the soft tissues than to the implant material. In an *in vitro* study, discoloration was more pronounced with Ti implants than with Zr implants, with an average vestibular soft tissue thickness of 1.68 mm (SD = 0.91) (*Thoma et al., 2016*). Therefore, in cases with a thin gingival biotype, especially in the aesthetic area, Zr implants seem to be a better choice.

This systematic review has a few limitations. First, this review only included RCTs, which may have a limited sample size and follow-up duration. However, we presented findings from cohort studies in the Discussion section to assess as many Zr implants as possible. Second, due to methodological limitations, we could not assess the effects of the implant manufacturer, site, and participant characteristics on outcomes. However, we assessed the potential influence of the implant site in the meta-regression analysis. We also stressed the importance of unanalyzed factors in the Discussion. We consider that these limitations may not compromise the validity of the results but may change our interpretation. Therefore, these findings should be evaluated, interpreted, and applied with caution.

## CONCLUSION

Zr implants may have a higher failure rate due to their mechanical weakness. Zr implants should be strictly assessed before they enter the market. Further studies are required to confirm these results.

## ACKNOWLEDGEMENTS

We appreciate Mr. Chunlin Qian for downloading the full texts of the articles, and Miss Ruiying Yin for assistance with software operations during the study. We would like to thank Editage for editing and reviewing this manuscript for English language.

### Funding
The authors received no funding for this work.

### Competing Interests
The authors declare there are no competing interests.

### Author Contributions
- Chengchen Duan conceived and designed the experiments, performed the experiments, analyzed the data, prepared figures and/or tables, authored or reviewed drafts of the article, and approved the final draft.
- Li Ye conceived and designed the experiments, performed the experiments, analyzed the data, authored or reviewed drafts of the article, and approved the final draft.
- Mengyun Zhang performed the experiments, analyzed the data, authored or reviewed drafts of the article, and approved the final draft.
- Lei Yang analyzed the data, authored or reviewed drafts of the article, and approved the final draft.
- Chunjie Li analyzed the data, authored or reviewed drafts of the article, and approved the final draft.
- Jian Pan analyzed the data, authored or reviewed drafts of the article, and approved the final draft.
- Yingying Wu conceived and designed the experiments, analyzed the data, authored or reviewed drafts of the article, and approved the final draft.
- Yubin Cao conceived and designed the experiments, performed the experiments, analyzed the data, prepared figures and/or tables, authored or reviewed drafts of the article, and approved the final draft.

### Data Availability
   The raw data used in the meta-analysis is available in Figure 2.

### Supplemental Information
Supplemental information for this article can be found online at http://dx.doi.org/10.7717/peerj.15010#supplemental-information.

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
