# Peer review of "Clinical performance of zirconium implants compared to titanium implants: a systematic review and meta-analysis of randomized controlled trials"

_PeerJ, doi:10.7717/peerj.15010_

## Round 0.1 · original submission · Minor Revisions

Please address the Reviewers' comments.

Reviewers 1 & 3 have requested that you cite specific references. You may add them if you believe they are especially relevant. However, I do not expect you to include these citations, and if you do not include them, this will not influence my decision.

Reviewer 1 ·

Basic reporting

A review report of the manuscript titled “Clinical performance of zirconium 1 implants compared to titanium implants: a systematic review and meta-analysis of randomized controlled trials”. Authors of current paper aimed to quantitatively assess and compare the clinical outcomes, including survival rate, success rate, and peri-implant indices of titanium and zirconium implants in randomized controlled trials. They concluded that zirconium implants may have higher failure rates due to their mechanical weakness.

Here are my concerns, questions and recommendations:
1. I could not understand why authors did not search also in Scopus database?
2. Although it is review paper, however in my view similarity index is very high. It is above 31%, so please reduce it up to reasonable limit.
3. Line 37. The sentence not clear.
4. For the Introduction I recommend the recent publication to include: Rokaya, D. et al. (2022). Metallic Biomaterials for Medical and Dental Prosthetic Applications. In: Jana, S., Jana, S. (eds) Functional Biomaterials. Springer, Singapore. https://doi.org/10.1007/978-981-16-7152-4_18
5. The searching strategy should be clarified further; which terms were used (Mesh)?

Experimental design

Please see above. It should be corrected according to comments.

Validity of the findings

Please see above. It should be corrected according to comments.

Additional comments

none

Reviewer 2 ·

Basic reporting

The article is written in a good and technically correct English, lierature references and article structure are valid and coherent within the scope of the review.

Experimental design

Methodology of the review is clearly stated. Wide description has been given to the search strategy, as well as to the most used systems for systematic reviews. Assessment of risk of bias appropriate.

Validity of the findings

The impact and novelty of this review is well stated and , atm y knowledge, may represent an important milestone in implant dentistry.

Reviewer 3 ·

Basic reporting

Dear Authors

This is very good work and extraction of the information in this paper regarding Clinical performance of zirconium implants compared to titanium implants. I would recommend to expand the introduction first paragraph. Line 53 till 60. The below references and book chapter will help authors to expand these lines related to titanium.
a) https://www.sciencedirect.com/science/article/pii/B9780128195864000020

b) https://doi.org/10.1563/aaid-joi-D-16-00072


Expand line 64-69 from same book chapter above mentioned.

Experimental design

In Materials & Methods, authors mentioned PRISMA 2015. Please do revise and use latest PRISMA 2020. For your convenience please see the weblink for creating the PRISMA2020 flow diagram and method. https://prisma-statement.org/prismastatement/flowdiagram.aspx

In Search strategy heading from line 99- 104 authors can add the search methodology reference e.g.;
https://doi.org/10.1016/j.jtumed.2021.05.012

Authors reported that they used The GRADE pro system in line 145. Why they not mentioned the reference source for this system. Please add.

Validity of the findings

Results section is written well.

---

## Round 0.2 · Major Revisions

Please address the comments from the reviewer regarding the PRISMA flow diagram.

Reviewer 3 ·

Basic reporting

Authors revised very well but my one serious concern not rectify from authors related to PRISMA 2020 flow diagram. They not revised the diagram only cited the reference.

Experimental design

Authors need to revise flow diagram.

Validity of the findings

Perfect.

---

## Round 0.3 · accepted · Accept

I have gone through the article and the comments. The authors have corrected the manuscript according to Reviews 1 and 2.

I think the manuscript can be accepted.